# Changes in IL-6, IL-12, IL-5, IL-10 and TGF-β1 Concentration in Patients with Treatment-Resistant Schizophrenia (TRS) Following Electroconvulsive Therapy (ECT)—A Pilot Study

**DOI:** 10.3390/biomedicines12112637

**Published:** 2024-11-19

**Authors:** Anna Maria Szota, Izabela Radajewska, Małgorzata Ćwiklińska-Jurkowska, Kinga Lis, Przemysław Grudzka, Wiktor Dróżdż

**Affiliations:** 1Department of Psychiatry, Ludwig Rydygier Collegium Medicum in Bydgoszcz, Nicolaus Copernicus University in Torun, Curie-Skłodowskiej Street 9, 85-094 Bydgoszcz, Poland; izabela.radajewska@cm.umk.pl (I.R.); przemyslaw.grudzka@cm.umk.pl (P.G.); wikdr@cm.umk.pl (W.D.); 2Department of Biostatistics and Biomedical Systems Theory, Ludwig Rydygier Collegium Medicum in Bydgoszcz, Nicolaus Copernicus University in Torun, Jagiellonska Street 13-15, 85-067 Bydgoszcz, Poland; mjurkowska@cm.umk.pl; 3Department of Alergology, Clinical Immunology and Internal Diseases, Ludwig Rydygier Collegium Medicum in Bydgoszcz, Nicolaus Copernicus University in Torun, Ujejskiego Street 75, 85-168 Bydgoszcz, Poland; kinga.lis@cm.umk.pl

**Keywords:** electroconvulsive therapy (ECT), schizophrenia, cytokines, interleukins, antipsychotic drugs

## Abstract

**Background/Objectives:** Treatment-resistant schizophrenia (TRS) may be considered as a neuro-immune disorder. Electroconvulsive therapy (ECT) remains an important therapeutic option for patients with TRS, however, its impact on cytokine profile is barely investigated. Therefore, this study attempts to establish associations between serum cytokines IL-6, IL-12, IL-5, IL-10 and TGF-β1 changes (pre- and post-ECT) and the effectiveness of ECT in TRS patients. The second aim is to search for correlations between serum concentrations of the above specified cytokines and psychometric assessments of clinical schizophrenia symptoms. **Methods:** The cytokine concentrations were measured in eight TRS patients on psychopharmacological treatment prior to and following ECT and in 13 control subjects. Psychopathology assessment was based on the Positive and Negative Syndrome Scale (PANSS). **Results**: Prior to ECT, IL-10 concentration was significantly higher in TRS patients, while IL-5 was decreased in comparison to the controls. A significant concentration decrease in the pro-inflammatory cytokines IL-6 (*p* = 0.012), IL-12 (*p* = 0.049) and anti-inflammatory IL-10 (*p* = 0.012) post-ECT vs. pre-ECT was observed, whereas concentrations of IL-5 and TGF-β1 did not significantly change. Also, a significant decrease in schizophrenia symptoms measured by the PANSS post-ECT was found. Furthermore, the pattern of correlations between PANSS scores and cytokine concentrations was different when comparing levels pre- and post-ECT. Additionally, correlations between changes in PANSS scores and cytokine concentrations were found. **Conclusions:** These results may indicate the probable impact of electroconvulsive therapy on the balance between pro- and anti-inflammatory cytokines, which may correspond to a neurobiological therapeutic effect of ECT in TRS patients.

## 1. Introduction

Schizophrenia (SCZ) is a chronic, debilitating psychotic disorder with heterogenous symptomatology, which includes positive symptoms (hallucinations and delusions), negative symptoms (social withdrawal, apathy, anhedonia and blunted affect) and cognitive dysfunction [1,2]. SCZ phenotypes include first-episode psychosis, acute schizophrenia episodes, chronic and deficit SCZ but also treatment-resistant schizophrenia (TRS) [3,4]. Regardless of the schizophrenia phenotype, incidence is associated with genetic predisposition [5], infection [6], autoimmune conditions [7] but also with activation of the immune system [2,3,4,8].

Treatment-resistant schizophrenia (TRS) is usually defined as a lack of response to at least two first-line antipsychotic drugs at an adequate dosage, appropriate treatment duration and confirmed compliance [9]. TRS affects 20–30% of patients with schizophrenia (SZ), resulting in poor daily functioning, reduced quality of life, increased numbers of hospitalization and a higher level of unemployment in comparison to treatment responders [10]. Development of TRS, at least in part, is caused by the deregulation of immune processes reflected by fluctuating levels of inflammatory molecules [11,12].

Studies conducted so far have revealed that TRS may be considered as a neuro-immune disorder characterized by an activated Inflammatory Response System (IRS), which is counterbalanced by a conversely activated Compensatory Immune Response System (CIRS). IRS activation refers to an increase in the production of pro-inflammatory cytokines by M1 macrophages (IL-1β, IL-6, and tumor necrosis factor-alpha [TNF-α]), Th-1 (IL-2, IL-12, and interferon-gamma [IFN-γ]) and Th-17 (IL-17) cells. In contrast, CIRS activation produces more anti-inflammatory factors, involving Th-2 (IL-4, IL-5) and T regulatory (Treg; IL-10 and TGF-β1) immune cells [4,13,14,15]. Increased IRS activation and overproduction of pro-inflammatory cytokines released by M1 and Th-1 have many adverse effects on neuronal cells, including oxidative damage to neurons, changes in neuroplasticity and neuronal signaling, and decreasing neurogenesis, which may be indicators of the neuroprogression of schizophrenia. Released cytokines may cross the blood–brain barrier (BBB), causing changes in the central inflammatory process (sustaining neuroinflammation) and activating astrocytes and microglia to produce cytokines [16,17]. Activated microglia secreting IL-1α, and TNF may cause neuronal and glial death [18] leading to the progressive loss of brain tissue in schizophrenia [19]. Moreover, activated Th-1 and M1 subsets stimulate the production of neurotoxic tryptophan catabolites (TRYCATs), including picolinic and quinolinic acids, which may cause cognitive impairments in TRS patients [20,21]. Increased production of IRS cytokines also has an impact on the induction of both negative symptoms of schizophrenia and PHEM (psychosis, hostility, excitation and mannerism) symptoms [3].

Simultaneously, pro-inflammatory cytokines such as IL-6, IL-1β and TNF-α promote liver production of acute-phase proteins, such as C-reactive protein (CRP), Hpx, alpha-1 antitrypsin, Hp, alpha 1S and alpha-2 macroglobulin, which, except for CRP, have significant anti-inflammatory effects. These proteins stimulate the production of anti-inflammatory IL-10 and heme-oxygenase-1 and attenuate production of pro-inflammatory IL-8 and IL-17 [22]. In addition, alpha-2 macroglobulin shows potent anti-inflammatory and antioxidant effects by binding to pro-inflammatory cytokines (IL-6, IL-1β and TNF-α) [23].

An exaggerated IRS response is counterbalanced by the activation of the CIRS system (anti-inflammatory cytokines production) and neurotrophic defenses (including antioxidants and brain-derived neurotrophic factor (BDNF) production), but this is relatively inadequate to inhibit IRS activation; therefore, impairments in the CIRS response are correlated with the severity of schizophrenia symptoms and poor clinical outcomes in TRS [4,14,24]. Even with remission of schizophrenia symptoms using antipsychotic treatment, both the IRS and CIRS remain overactive, suggesting a new homeostatic set-point between both components [4,25].

Currently available studies have revealed that TRS is associated with elevated levels of some pro- and anti-inflammatory cytokines, including interleukins such as IL-2, IL-6, IL-12, IL-10, IL-17A, beta2 microglobulin (B2M), IL-6 receptor (IL-6R), IL-1R antagonist (IL-1RA), interferon γ (IFN-γ), macrophage inflammatory protein 1α (MIP-1α) and chemokines (CCL2, CCL3, CCL11), but the results are inconsistent, so further research in this area is required [4,8,11,26,27].

The recommended option of treatment for TRS patients is electroconvulsive therapy (ECT) [28,29,30]. A recent meta-analysis has shown the efficacy of ECT in the improvement of global and positive symptoms of schizophrenia (10 studies, effect size = 0.45), but not in negative symptoms, in comparison to continuation of antipsychotic treatment [31]. The neurobiological mechanisms leading to the therapeutic effects of ECT are not yet fully understood, but research has revealed the involvement of the acceleration of neural network activity, neurogenesis promotion, neural hyperconnectivity, normalization of the activity of hormonal axis (i.e., the hypothalamus–pituitary–adrenal [HPA]) and the hypothalamus–pituitary–thyroid [HPT]) and also a reduction in the neuroinflammatory state [32].

To the best of our knowledge, there are just a few studies that focus on the impact of ECT on the immune system in TRS patients. The research conducted so far in TRS patients looks at molecules such as brain-derived neurotrophic factor (BDNF), vascular endothelial growth factor (VEGF), TNF-α, IL-4, transforming growth factor β1 (TGF-β1), myeloperoxidase (MPO) and nuclear factor-kB (NF-kB). It has been found that ECT may have a twofold influence on the concentration of BNDF and VEGF. Some studies have documented a significant increase in BDNF and VEGF serum levels associated with the therapeutic action of ECT [33,34,35,36], whereas other studies have reported no changes in BDNF and VEGF post ECT vs. pre-ECT [37,38,39,40]. Kartalci et al., 2016 [41] reported a significant increase in serum concentrations of IL-4 and TGF-β1 post-ECT in comparison to pre-ECT, and also a correlation between this effect and a clinical improvement (decrease of schizophrenia symptoms) in TRS patients. Also, MPO and NF-kB concentrations were unaffected by ECT [41]. In the case of TNF-α, a decrease post-ECT v. pre-ECT was noticed [40]. Additionally, regarding BDNF, VEGF and TNF-α, there were no significant concentration differences between TRS patients and control subjects [40].

Bearing in mind that TRS may be considered a neuro-immune disorder and is based on the scarce, aforementioned research on the involvement of particular cytokines in TRS development and their changes following ECT, our pioneering study is justified.

Therefore, this study attempts to establish associations between serum cytokines IL-6, IL-12, IL-5, IL-10 and TGF-β1 changes and the effectiveness of ECT in treatment-resistant schizophrenia patients. The second aim is to search for correlations between serum concentrations of the above specified cytokines and psychometric assessments of clinical schizophrenia symptoms performed with the Positive and Negative Syndrome Scale (PANSS).

## 2. Materials and Methods

### 2.1. Patients, Inclusion and Exclusion Criteria

This study was conducted at the Psychiatry Clinic of the University Hospital no 1 in Bydgoszcz (Poland). Participants were inpatients with treatment-resistant schizophrenia referred for ECT by their psychiatrist. The recruitment period lasted from December 2018 to February 2020 but was untimely finished because of the COVID-19 pandemic. These patients typically failed to achieve a clinical response to at least three separate trials of antipsychotic medication at a sufficient dose. The number of ECT sessions to be administered was based on the patient’s clinical response. A total of 8 patients (5 men and 3 women, mean age 35) were recruited. A detailed description of included patients is given in Table 1.

The study inclusion criteria comprised patients aged 18–65 years diagnosed with schizophrenia (paranoid or residual) according to the ICD-10, who fulfilled the criteria of drug resistance. Exclusion criteria from the study were serious neurological diseases, current addiction to alcohol or psychoactive substances, autoimmune diseases, lack of informed consent to electroconvulsive therapy and contraindications to conduct ECT such as myocardial infarction within the last 3 months, decompensated heart failure, severe valvular heart defects, unstable angina, complex arrhythmias, aortic aneurysm, uncontrolled diabetes mellitus, decompensated renal failure, severe metabolic disorders, severe lung disease, acute glaucoma attack, stroke within the last 4 weeks, intracranial neoplasm and retinal detachment.

A total of 13 healthy subjects (2 men and 11 women, mean age 33 years) with no history of past or current mental disorders were recruited as a control group. After the procedures were fully explained, written informed consent was obtained from all subjects to participate in the study. The current study was conducted in accordance with the Declaration of Helsinki and approved by the Ethics Committee of Nicolaus Copernicus University in Toruń Collegium Medicum in Bydgoszcz, Poland. Approval number KB 631/2018, obtained on 24 October 2018.

The male/female ratio was 5/3. Patients were aged 20–41 years, with the mean being 35 years. The length of disease in the group was between 3 and 23 years, with the mean being 13.9 years and median being 16 years. Two patients were diagnosed with residual schizophrenia and six with paranoid type according to the ICD-10 criteria. Only one individual had been previously treated with ECT. Three patients had history of suicidal attempt/s. Five subjects were diagnosed with past or current SUD; however, two of them had nicotinism only. All patients were treated with at least two antipsychotics and satisfied criteria of treatment-resistant schizophrenia [9,42]. Moreover, five subjects met criteria for ultra-treatment-resistant schizophrenia [9,43] and all of them were treated with higher chlorpromazine equivalent antipsychotic dosage as compared to those not treated with clozapine. Four patients took a remarkably high combined antipsychotic dose expressed in chlorpromazine equivalent (>1000 mg a day). The mean antipsychotics’ dose calculated as chlorpromazine equivalent for the group was 1108 mg a day and median 913 mg a day [44]. This suggests the severity of disease. The mean post-ECT PANSS score improvement in the group on all subscales and in total was at least 30% and may be regarded satisfactory against treatment-resistance. These 3 individuals were continuing the phramacological treatment, so it should be suitable for treatment. https://psychopharmacopeia.com/antipsychotic_conversion.php; accessed on 10 June 2024. Antipsychotic Dose Conversion Calculator.

### 2.2. ECT Treatment Procedures

Before undergoing ECT, each patient was screened for general health through a physical and neurological examination, blood and urine tests, electrocardiogram and a cerebral computed tomography scan. The severity of schizophrenia symptoms was evaluated twice by conducting the PANSS, before ECT and after the last session of ECT. Each patient underwent 3 sessions of ECT per week for a mean number of 13 ECT sessions (range: 11–15). Only one patient had a history of ECT. A brief-pulse machine generating bipolar square waves was used (Thymatron System IV, Somatics; Lake Bluff, IL, USA). Electrodes were positioned bilaterally on the frontal–temporal region. Only one adequate seizure was required for each session, which was defined as an electroencephalographic seizure lasting more than 20 s with a high-amplitude, slow wave and postictal suppression. The electrical dose was maintained at the threshold level unless short seizures occurred, in which case the charge was increased by 50%. Anesthesia was induced with i.v. thiopental (3–4 mg/kg, i.v.) and succinylcholine (0.5–1.5 mg/kg, i.v.). The total number of ECT sessions for each patient was determined by the consensus opinion of the medical staff based on clinical improvement and adverse effects.

### 2.3. Interleukins Assay

For cytokine analysis, venous blood samples were collected from all of the patients with schizophrenia between 7.00 and 7.30 a.m. pre-ECT and after the last ECT session. The blood samples were drawn into anticoagulant free tubes and after 45 min, serum was separated by centrifugation at 3000 rpm for 10 min at 4 °C. Then, the serum samples were pipetted into sterile 1.5 mL microcentrifuge tubes and stored at −80 °C until assay. Also, the venous blood was collected once on a screening day from healthy people who comprised the control group. All samples were analyzed using ELISA kits (Diaclone, Medics Biochemica Group, Besançon, France) according to the manufacturer’s protocol. Cytokine levels are expressed as pg/mL.

### 2.4. Statistical Methods

To support the limited number of patients, special statistical analysis was performed. To improve the power of the tests, the values were logarithmized to obtain normal distributions. Additionally, to avoid the drawbacks of the relatively small samples and to increase the power, Bayesian analyses were applied. Bayesian related-sample inference was employed for comparison of the cytokines before and after ECT. For the testing of any differences between distributions of results pre- and post-ECT, a Bayesian Student *t*-test was applied. Similarly, Bayesian related-sample inference was applied for comparison of the PANSS scores before and after ECT. Additionally, Bayesian independent sample inference was applied for testing the hypotheses concerning the comparison of cytokine concentrations in patients before ECT with controls. Also, Bayesian independent sample inference was applied for testing the hypotheses concerning comparison of cytokine concentrations in patients following ECT with controls. For the testing of differences between distributions of results pre-ECT with the control group or post-ECT with the control group, Bayesian *t*-tests for independent variables were applied. Correlations were calculated with Spearman statistics and a scatterplot for two-dimensional distributions is presented with fitted lines and determination coefficients. PS IMAGO PRO v. 9.0 package was used for calculations https://en.predictivesolutions.pl/en/ps-imago-pro; Accessed on 10 October 2024. 

## 3. Results

### 3.1. Changes in Cytokine Concentration and PANSS Score Following ECT

After logarithmization, normal distributions for both cytokines and PANSS scores were obtained, according to Shapiro–Wilk tests on the 0.05 levels. Thus, Bayesian inference for normal distributions was applied [45,46,47]. A significant decrease in the PANSS positive symptoms, PANSS negative symptoms, PANSS general symptoms and PANSS total score in schizophrenic patients was observed (Table 2). This reflects substantial improvement in psychopathology following ECT.

The results of cytokine concentrations are presented in Table 3 with means and standard deviations (SDs). According to the *p*-values of the Shapiro–Wilk test after logarithmization, both pre-ECT and post-ECT results were compared, and Bayesian *t*-tests for dependent variables were employed. Similarly, for comparisons to the control group, the Shapiro–Wilk test did not reject the hypothesis of normality in both compared groups. According to the *p*-value calculated on this, a Bayesian *t*-test for unpaired samples was employed.

We observed significant changes post-ECT vs. pre-ECT for pro-inflammatory cytokines: IL-6, IL-12 and anti-inflammatory IL-10 and all medians decreased (Table 3). When comparing pre-ECT cytokine concentrations with the control group, significant differences were found for IL-10. When comparing post-ECT cytokine concentrations with the control group, significant differences were observed for IL-6.

### 3.2. Correlations Between Cytokines and PANSS Scores During ECT

Correlations r are given in the table with marked significant values, according to the calculated probability of >|r| under the assumption of the null hypothesis of zero correlation.

The Spearman correlations were calculated for cytokines and the PANSS scores before ECT (lower triangle in Table 4) and following ECT (upper triangle in Table 4). There were significant positive correlations between the scores of the PANSS subscales and the PANSS total scores before ECT, but only one significant correlation 0.882 following ECT.

All corresponding PANSS subscales or total scale before and following ECT were highly positively correlated (diagonal values in Table 4). Moreover, we found a highly significant negative correlation between TGF-β1 and the PANSS positive symptoms following ECT (r = −0.790, *p* = 0.02). This means that a higher intensity of positive symptoms on the PANSS at ECT completion was related to smaller TGF-β1 values in TRS patients.

The only significant positive correlation between cytokine concentrations in the control group was found for TGF-β1 and IL-6 (Table 5).

Correlations between cytokine concentration changes, changes on the PANSS subscales and total score following ECT (columns) with variables referring to both cytokine concentrations and psychometric assessments (rows) were calculated for patients and the results are presented in Table 6 and Table 7. Significant correlations have been marked.

A higher concentration of pre-ECT IL-12 correlated positively (0.93) with a significant decrease in IL-12 post-ECT. Greater intensity of positive symptoms on the PANSS before ECT was positively associated (r = 0.71) with a significant decrease in the score on the PANSS positive subscale as a result of ECT. Higher scores on the PANSS negative symptoms subscales before ECT were positively correlated with a greater decrease in the total PANSS score following ECT (0.84). Both general symptom intensity and total score of the pre-ECT PANSS were associated positively with a more pronounced decrease in the PANSS total score post-ECT (0.85 and 0.85).

A higher IL-12 concentration following ECT was positively correlated with a greater decrease in both negative symptoms score and total score of the PANNS (r = 0.87 and 0.74, respectively). A lower post-ECT IL-5 concentration was associated with a greater decrease in IL-5 as a consequence of ECT (−0.98). Similarly, lower concentration of TGF-β1 following ECT correlated with both IL-6 and TGF-β1 decreased due to ECT (r = −0.76 and −0.93, respectively; Table 6).

The results of correlations between changes in cytokine concentrations and changes in the PANSS scores pre- to post-ECT indicate that the decrease in total PANSS score was positively correlated with a decreased intensity of negative and general symptoms (r = 0.77 and 0.81). Thus, patients with a higher decrease in total PANSS score also had a higher decrease in negative and general symptoms. The only significant correlation among cytokines was a positive association between changes in IL-6 and TGF-β1 concentrations pre- to post-ECT (0.83, Table 7). Thus, patients with a higher decrease or increase in IL-6 concentrations also had a higher decrease or increase (respectively) in concentrations of TGF-β1. Interestingly, the same correlation was observed in the control group, i.e., a positive association of IL-6 and TGF-β1 concentrations (0.78, Table 5).

Figure 1 presents changes in IL-6 and TGF-β1 concentrations following ECT for each patient. Post-ECT values (triangles) are lower than pre-ECT (circles); however, there is no linear association between these parameters. Conversely, an association between IL-6 and TGF-β1 concentrations in the control group is linear; a corresponding scatterplot for *n* = 13 control subjects on Figure 2 shows the determination coefficient R^2^ = 0.60 and a higher Spearman correlation (r = 0.78; *p* = 0.002) than in schizophrenic patients (both pre- and post-ECT) (compare Figure 1).

Figure 3 shows a scatterplot of the differences of post-ECT from pre-ECT values in IL-6 and TGF-β1, after subtracting coordinates in the horizontal and vertical axes of red and blue points in Figure 1. TGF-β1 and IL-6 decreases are significantly dependent with the Spearman correlation at 0.83 (*p* = 0.01). The overlaid parabolic fit line depicts a nonlinear dependence with determination coefficient R^2^ = 0.789.

In summary, the statistical analysis revealed the following: (a) a significant concentration decrease in pro-inflammatory cytokines IL-6 (*p* = 0.012), IL-12 (*p* = 0.049) and anti-inflammatory IL-10 (*p* = 0.012) post-ECT vs. pre-ECT; (b) no significant change in IL-5 and TGF-β1 concentration post-ECT vs. pre-ECT; (c) a significant decrease (*p* < 0.001) in schizophrenia symptoms (a significant decrease in all subscales: positive, negative, general PANSS) and total PANSS score led by ECT; (d) a negative correlation between TGF-β1 and PANSS positive symptoms (r = −0.790, *p* = 0.05) post-ECT.

## 4. Discussion

The main objective of this study was to assess the effects of ECT on cytokines such as IL-6, IL-12, IL-5, IL-10 and TGF-β1 in patients with treatment-resistant schizophrenia. The second aim was to determine correlations between serum levels of the above cytokines and changes in clinical schizophrenia symptoms measured by the Positive and Negative Syndrome Scale (PANSS).

To the best of our knowledge, the impact of ECT on the concentration of IL-6, IL-12, IL-5 and IL-10 in TRS patients has not yet been the subject of any research conducted so far. Changes in other cytokines, interleukins and neurotrophins post-ECT vs. pre-ECT in TRS patients are summarized in Table 8 together with the results of this study.

Our study shows that there were no statistically significant differences in blood concentrations of IL-6, IL-12 and TGF-β1 between TRS patients prior to ECT and the control group. However, a decreased concentration of IL-5 and an increased concentration of IL-10 (*p* < 0.001) in TRS patients in comparison to the control group was observed. Our findings on IL-6, IL-12 and TGF-β1 are in opposition to previous results, published by other authors, where increased (IL-6 and IL-12) [8,25,48] or decreased concentrations of IL-12 and TGF-β1 [25,41] in TRS patients in comparison to the control group were detected. Also, with regard to the concentration of IL-5 and IL-10 in TRS patients, the results are conflicting. Leboyer et al., 2021 [8], showed no significant differences in IL-5 and an increased concentration of IL-10 in TRS patients as compared to the control group, whereas Chen et al., 2023 [25], showed a decreased concentration of IL-10 in TRS patients. An explanation for such inconsistent results seems to be difficult due to limited evidence resulting from small sample sizes and the heterogeneous population of TRS patients, and the still not adequately understood inflammatory mechanisms underlying drug-resistance in patients with schizophrenia.

It has been suggested that both IL-6 and TGF-β1 are indicators of acute exacerbations of schizophrenia [49], so their levels are potentially increased during acute states in comparison to the control group [4]. We may only hypothesize that a lack of significant difference in IL-6 and TGF-β1 in TRS patients prior to ECT, in comparison to the control group, may result from a stable mental state in chronic schizophrenia, and not with acute exacerbation of the illness. Also, increased concentration of IL-10 (*p* < 0.001) in our TRS patients may inhibit production of both IL-6 and IL-12, which means their concentrations were not different from the control group. On the other hand, some proteins such as alpha-1 antitrypsin, Hp, alpha 1S and alpha-2 macroglobulin produced by the liver may stimulate the production of IL-10 [22], which exerts an anti-inflammatory effect, as was observed in our TRS patients. In addition, its binding to IL-6 alpha-2 macroglobulin [23] may be a reason why the IL-6 level is not increased in TRS patients. Moreover, our patients were on various combinations of antipsychotic drugs including haloperidol, olanzapine, quetiapine, aripiprazole, clozapine and mood stabilizers; therefore, their possible effect on the immune system has to be taken into account. In vivo research showed that typical antipsychotics suppress plasma IL-6 and IL6R [13], whereas repeated administration of atypical antipsychotics, i.e., clozapine or risperidone, may have a dual effect. Significantly increased plasma concentrations of IL-2R, IL-6 and TNF-α in TRS patients treated with risperidone were found by Maes et al., 1997 [13], Maes et al., 1994 [50], and Tourjman et al., 2013 [51]. Meanwhile, Patlola et al., 2023 [52], in a meta-analysis observed decreased levels of IL-6 and TNF-α after treatment with risperidone vs. before treatment in patients with chronic schizophrenia. With regard to clozapine treatment, there were no significant differences in the concentration of IL-6 and TNF-α after treatment vs. before treatment in patients with chronic schizophrenia [52]. Also, aripiprazole may decrease IL-12 levels in patients with chronic schizophrenia, or may increase IL-10 (*p* < 0.001) levels when used over the long term [53]. Increased IL-10 levels have also been reported in patients with chronic schizophrenia treated with risperidone or clozapine for at least 6 weeks [54]. Therefore, various combinations of antipsychotics may be involved in the lack of differences in IL-6 and IL-12 concentrations as well as in the higher IL-10 concentration in TRS patients pre-ECT in comparison to the controls. Moreover, no differences in the concentration of IL-6 and IL-12 in TRS patients prior to ECT vs. the control group may suggest that IL-6 is a state marker, and simultaneously undermines IL-12 status as a trait marker. Therefore, IL-6 concentration reflects the intensity of symptoms in TRS patients with a tendency for normalization under antipsychotic treatment. A significant increase in IL-6 concentration was found in multiple-episode schizophrenia (MES) patients [55,56]. Antipsychotic treatment in our TRS patients probably decreased IL-6 concentration to similar values as in the control subjects. On the other hand, IL-12 elevation as a trait marker represents the properties of the behavioral and biological processes that play an antecedent, possibly causal, role in the pathophysiology of schizophrenia [56] and is present in acutely relapsed patients and after antipsychotic treatment [55]. Consistently, IL-12 should be higher in our patients on antipsychotic treatment as compared to the control group. However, we did not observe such an effect in our study, so this suggests that IL-12 may not be a trait marker for TRS. More pronounced cytokine alterations have been found to predict a lack of response to antipsychotic treatment and also unfavorable long-term outcomes in schizophrenia [57].

The latest reviews and meta-analyses have ascertained the impact of various combinations of antipsychotic medications on interleukin concentrations in patients with chronic schizophrenia, but not TRS, and they have given unequivocal results [52,58,59]. There are no data on the impact of combinations of antipsychotics in TRS patients on other cytokines, e.g., IL-2, IL-4, IL-5, IL-10, IL-12, IL-17, IL-23 and TGF-β1, although their involvement in schizophrenia pathophysiology has been demonstrated [58,60]. With this in mind, the results of our study may be perceived as pioneering.

In our study, correlations between serum levels of cytokines and the PANSS scores pre- and post-ECT were evaluated in TRS patients. No correlation was found between any of the measured cytokines (IL-6, IL-12, IL-5, IL-10, TGF-β1) and positive, negative, general or total PANSS scores pre-ECT. The only available data that our results can be compared with were published by Dahan et al., 2018 [61]. They found a positive, statistically significant correlation between symptom severity measured by the PANSS and IL-6 (r = 0.482, *p* = 0.0039 with the overall score; r = 0.466, *p* = 0.0055, with the general score; r = 0.350, *p* = 0.0425, with the negative scale) [61]. Furthermore, Noto et al., 2015 [11], found correlations between the PANSS negative score and IL-2 (*p* = 0.021, negative association) and CCL11 (*p* = 0.005, positive association) in TRS patients.

Many more correlations between inflammatory markers and the different types of symptoms in schizophrenia were found in first-episode and drug-naïve schizophrenia (FEDN), and in patients with chronic schizophrenia. It was found that higher levels of IL-6, IL-1β, IL-33 and IL-17 are associated with more severe positive symptoms [17,58,62,63]. Exacerbated negative symptoms are positively correlated with increased levels of IL-6, IL-1β, IL-4, IL-8, TNF-β, TGF-β1 and IL-13 [17,57,64]; however, the correlation between IL-1β and TNF-α and negative symptoms is only seen in chronic patients [58]. A negative correlation with negative symptoms was found for IL-2 and IL-10 [58]. Also, increased levels of IL-6, IL-33 and TGF-β1 are positively correlated with the general psychopathology sub-score [54,58]. The total PANSS score is positively correlated with the levels of IL-6, IL1β, IFN-γ, IL-17 and TGF-β1 [61,65,66].

It was suggested that on the basis of the correlation between cytokine concentrations and PANSS scores before treatment, it is possible to predict a therapeutic effect of the treatment in some distant future (e.g., at 6-month follow-up). A good example is results obtained by He et al., 2020 [2], in patients with a first episode or relapse of schizophrenia. They found that a higher IL-6 level (*p* = 0.027) and lower IL-8 level (*p* = 0.035) predicted a better therapeutic effect of antipsychotic treatment on negative symptoms. A higher IL-6 concentration also predicted less improvement in depressive symptoms of schizophrenia [2]. As in our results, there was no correlation between any of the measured cytokines (IL-6, IL-12, IL-5, IL-10, TGF-β1) and positive, negative, general or total PANSS scores pre-ECT. A recently published study performed in male schizophrenic patients has revealed that catalase and IL-6 serum elevation could be predictors of treatment-resistance [67]. Other studies in this field were conducted in non-TRS samples [58,68,69,70]. Thus, predictions of a therapeutic effect in TRS using cytokine profiles require further research [68].

The results of our study revealed a significant decrease in plasma concentrations of IL-6 (*p* = 0.012), IL-12 (*p* = 0.049) and IL-10 (*p* = 0.012) in schizophrenic patients after ECT in comparison to pre-ECT. Also, a significant decrease in the concentration of IL-6 (*p* = 0.005) and IL-12 (*p* = 0.013) after ECT in comparison to the control group was found. With regard to IL-5 and TGF-β1, the change after ECT in comparison to pre-ECT was not significant. To date, studies examining the effect of ECT on the same inflammatory mediators we considered in TRS patients are not available, except Kartalci et al., 2016 [41], who showed an increased concentration of TGF-β1(*p* = 0.001) following ECT. Our results seem to be partly in line with those obtained by Kartalci et al., 2016 [41], as the concentration of TGF-β1 was increased following ECT, although without reaching statistical significance.

TGF-β1 modulates anti-inflammatory and repairing processes [71]; thus, its elevation subsequent to ECT seems reasonable. The positive correlation between IL-6 and TGF-β1 observed in our study is in accordance with the findings of Zhang et al., 2005 [71], which may point to a dynamic interplay of both cytokines and their influence on inflammation and healing processes through regulatory T cells (Treg) [71]. This may suggest the ECT stimulation of both pro- and anti-inflammatory systems in the brain that may promote the shaping of a new type of balance between them [64].

In the case of interleukins such as IL-6, IL-12, IL-5, IL-10 and PANSSs (in all subscales and total PANSS), no correlation was found following ECT. The only significant negative correlation that was noticed was between TGF-β1 and PANSS positive symptoms (r = −0.790, *p* = 0.05) after ECT. This may indicate that the TGF-β1 concentration increase induced by ECT could be coupled with the attenuation of positive symptoms measured by PANSSs. Interestingly, Kartalci et al., 2016 [41], found that TGF-β1 elevation was negatively correlated with the changes in Brief Psychiatric Rating Scale (BPRS) scores following ECT (*p* = 0.04). This may suggest a connection of TGF-β1 anti-inflammatory and repairing properties to the effect of ECT on psychopathology in treatment-resistant schizophrenia.

Not surprisingly, we observed significant improvements on the PANSS positive, negative, general and also total scores following ECT. The differences in medians were −9.5; −11.5; −18; and −41, respectively. Previously published studies in TRS patients in which various combinations of antipsychotics, including clozapine, along with ECT were used also noticed such an effect [64,72,73,74,75,76]. The PANSS score reductions observed in these studies were 40–71% (on all subscales and overall score) and a 40% decrease on the BPRS score, regardless of medication(s) dose(s), years of treatment and number of ECT sessions. Moreover, s meta-analysis including 23 studies found the superior efficacy of ECT augmentation of clozapine (nine studies) compared to ECT augmentation of non-clozapine (typical and atypical) antipsychotics [77]. The majority of the patients enrolled in our study (five out of eight) were on clozapine and therefore a significant decrease in the PANSS scores as a result of ECT could be regarded as both expected and consistent with evidence.

There is a paucity of high-quality data on brain ECT changes in schizophrenia since the majority of studies are based on small sample sizes of patients continuously treated with antipsychotics and, additionally, they have an observational design. However, the existing evidence indicates a multi-directional and simultaneous influence of this procedure on multiple systems. Among the most important modifications identified are those pertaining to brain morphology and activity as well as hormonal and immune system function. ECT produces intraneuronal (epigenetic) and synaptic remodeling that results in complex and inter-correlated biochemical, structural, neurophysiological, hormonal and immunological transformations [78,79,80]. Therefore, immune changes associated with ECT in schizophrenia may be perceived as an integral part of the therapeutic effect of this treatment. ECT-induced increase in BDNF, which is linked to neuroplasticity, has been considered to be a predictor of positive clinical outcomes [80]. ECT generates transient immune system activation but ultimately leads to attenuation of both peripheral inflammation and neuroinflammation. Wang and Zhang, (2024), revealed that repeated ECT decreases the number of neutrophils and the neutrophils to leukocytes ratio in patients with schizophrenia; i.e., peripheral inflammation was significantly reduced following ECT [81]. On the other hand, the ECT effect on anti-inflammatory cytokines has not been hitherto well studied [82]. Rojas et al., 2022 [78], proposed a hypothetical model of non-specific ECT influence on the immune system in psychiatric disorders in which reduction in the pro-inflammatory (IL-6, TNF-α) and increase in anti-inflammatory (IL-4, TGF-β1) cytokine levels is paralleled in the CNS immune systems’ function, i.e., diminution of microglia and astrocyte activity [78]. It has not been proven whether this model is adequate for schizophrenia.

With regard to TRS patients, data suggest that ECT may decrease the severity of schizophrenia symptoms, not only through changing the activity of the immune system, but also through the neurotropic system. It was found that increased levels of both BDNF and VEGF induced by ECT in TRS patients post-ECT vs. pre-ECT were positively correlated with improvement of clinical symptoms (significant reduction in the PANSS positive, negative and total scales) [33,35,36,40]. Moreover, increased BDNF levels may enhance dopamine synthesis, its turnover and the functioning of the dopaminergic system, which are disrupted in TRS patients [35]. Also, both BDNF and VEGF have been proven to reversibly modulate hippocampal synaptic plasticity and to improve hippocampal activity related to learning and memory [36]. Thus, these neurotrophins may indirectly be involved in the improvement of cognitive function in TRS patients after ECT. Furthermore, ECT increases blood–brain barrier permeability, which allows medications to enter the brain, accumulate in higher amounts and express their therapeutic effect (decrease symptoms of schizophrenia) [83]. Another possible mechanism by which ECT may influence the symptoms of schizophrenia in TRS patients is connected with ECT-induced changes in gray matter volume [84], and with microstructural changes in the limbic system [85,86]. Kawashima et al., 2023 [84], found that a regional gray matter volume (GMV) increase in the left pregenual anterior cingulate cortex was significantly correlated with percentage changes on the BPRS. Also, microstructural changes in key limbic structures (the left hippocampus and the right amygdala), observed in the ECT group, but not in the medication-only group, and hypothetically reflected by MRI texture, are associated with a clinical response to ECT in psychosis [85]. This confirmed the result of an earlier study conducted by Yang et al., 2020 [86], in which a decrease in functional connectivity post-ECT compared to pre-treatment between the right amygdala and the left hippocampus correlated positively with a percentage reduction in the PANSS total score. These findings support the neuroplasticity hypothesis of ECT and highlight the hippocampus and amygdala as potential targets for neuromodulation in psychosis [85].

To this end, we have found a statistically significant decrease in concentrations of pro-inflammatory cytokines IL-6 (*p* = 0.012), IL-12 (*p* = 0.049) after ECT in comparison to pre-ECT, and no significant difference in IL-5 and TGF-β1 concentration after ECT was noticed. Kartalci et al., 2016 [41], showed that anti-inflammatory cytokines TGF-β1 and IL-4 were increased after ECT and proposed that this could balance the inflammatory state in patients with schizophrenia. The mechanism and meaning of such an increase in IL-4 and TGF-β1 serum concentrations are not clear. Although the increase in serum TGF-β1 and IL-4 may be caused by electrical stimulation alone, this may also be a secondary phenomenon related to the effects of neurotransmitters or neuroendocrine mediators [41]. TGF-β1 also inhibits production of pro-inflammatory cytokines produced by macrophages and Th-1 cells (IL-1β, IL-6, IL-12, TNF-α, IFN, IL-13) and Th-17 cells (IL-17), and stimulates the production of sIL-1RA, which exerts an anti-inflammatory effect. Therefore, it may be hypothesized that reductions in serum IL-6 and IL-12 that were observed after ECT, at least in part, could be the result of increased production of TGF-β1, as this cytokine inhibits IL-6 and IL-12 synthesis. Also, IL-5 (produced by Th2 cells) and IL-10 (produced by Treg cells) may have immune-regulatory effects and with TGF-β1 may exert an anti-inflammatory effect (CIRS). IL-10 enhances the release of IL-1RA from macrophages and suppresses Th-1, M1, dendritic and cytotoxic cells, as well as B and natural killer (NK) cells, whereas IL-5 is a growth factor for B cells and eosinophils. However, according to some authors, IL-5 plays a role in neuroinflammation by increasing the activation and proliferation of microglial cells and has an adverse effect on brain neuronal functioning and neuroprotective processes [63,87].

There is evidence that ECT-induced improvement in schizophrenia psychopathology has different neurobiological correlates than improvement in the result of solitary neuroleptic therapy. Jiang et al., 2019 [88], observed bilateral hippocampal volume increase in a group of 21 schizophrenic patients treated with ECT, in both responders and non-responders, and this was not present in a similar group undergoing pharmacological therapy with antipsychotics. Moreover, increased functional connectivity between the hippocampus and brain networks associated with cognitive function was shown only in the ECT-responders [88]. Assessment of insula volume in 21 schizophrenic patients treated with ECT and antipsychotics revealed an increase in posterior regions, which was associated with symptomatology reduction. Conversely, in the control group of patients treated with antipsychotics only, a reduction in volume in posterior insula areas was observed [89]. Another study found increased functional connectivity between the thalamus and putamen in schizophrenic patients following 4-week ECT that was not present in patients treated with antipsychotics [90]. Also, different patterns of function of cerebral–cerebellar loops in schizophrenic patients who clinically improved after antipsychotic treatment only, as compared to patients following ECT, was described [91]. Two studies of default mode network (DMN) density in 21 schizophrenic patients after four weeks of combined ECT plus antipsychotic therapy found increased global functional connectivity within the DMN, which was associated with symptom improvement, and this effect was not seen in patients treated with neuroleptics [92]. Gray matter volume in four studied brain regions following 4-week ECT in 21 schizophrenic patients was increased and this may indicate brain plasticity induction by ECT, which was not observed in patients on antipsychotic therapy. Positive symptom reduction was linked to gray matter increases in limbic areas (i.e., parahippocampal gyrus and hippocampus) [93]. These results suggest a more complex and multimodal action of ECT in schizophrenia in comparison to antipsychotic treatment.

## 5. Conclusions and Limitations

This study reveals a significant decrease in the PANSS positive symptoms, PANSS negative symptoms, PANSS general symptoms and PANSS total score in TRS patients, which reflects a substantial improvement in psychopathology following ECT. However, because no correlation between any of the measured cytokines (IL-6, IL-12, IL-5, IL-10, TGF-β1) and positive, negative, general or total PANSS scores pre-ECT was found in the presented data, making any prediction about the therapeutic effect of the treatment via correlations with cytokine changes in TRS patients seems to be currently impossible and requires further research. Moreover, comparison of IL-6, IL-12 and IL-10 concentration before and after ECT revealed a significant reduction in the concentration of these cytokines, whereas IL-5 and TGF-β1 were not changed in TRS patients. The presented data also indicate that the lack of IL-6 and IL-12 concentration differences in TRS patients prior to ECT as compared to the control group may suggest that IL-6 is a state marker, whereas IL-12 seems not to be a trait marker in TRS patients. However, there is sufficient clinical and neurobiological evidence to support the efficacy of ECT in TRS patients; nonetheless, data on the immune system involvement in therapeutic processes are still scarce and therefore it is impossible to detect a meticulous mechanism of action of this treatment.

There are some limitations to our study. Mainly, the study was heterogenous and featured a small sample of TRS patients. Factors such as differentiated diet, smoking habits, psychological stress and lifestyle may have affected cytokine profiles [94,95,96,97]. Secondly, all patients had been treated with different combinations of medications, including antipsychotics, mood stabilizers and antidepressants before drug-resistance developed, so immune balance (activation of both IRS and CIRS systems) was already affected by pharmacotherapy. Additionally, all patients were on psychopharmacological treatment during ECT, and although doses were decreased, this probably modified both the expression and magnitude of inflammatory markers released into blood. However, conducting ECT in TRS patients without psychotropic treatment may not be possible due to ethical issues. Moreover, the induction of anesthesia and administration of muscle relaxants prior to the administration of ECT may also change the concentration of cytokines, for example, by reducing IL-1β and TNF-α levels [98]. Another issue is the blood collection time-point. Blood samples for our study were collected in the morning between 7.00 and 7.30 the day before beginning ECT and on the day after the last course of ECT. There is evidence of both acute and long-term effects of ECT on inflammatory parameters [99]; therefore, our blood collection time-point may be not optimal to detect all cytokine changes associated with ECT. Optimization of the time-point for cytokines measurements requires complete time-course studies, which have not been conducted so far. Also, ELISA kits used to assay cytokine levels were manufactured by different companies, so the quality of antibodies used in ELISA kits, the kit manufacture, and the operator’s skills may have affected the measurements [100]. Lastly, comorbidity, diet, alcohol consumption, sex and menstrual cycle should be controlled as they all affect the concentration of immunological markers. In sum, more comprehensive research including the issues mentioned above in larger groups of patients is recommended.

## Figures and Tables

**Figure 1 biomedicines-12-02637-f001:**
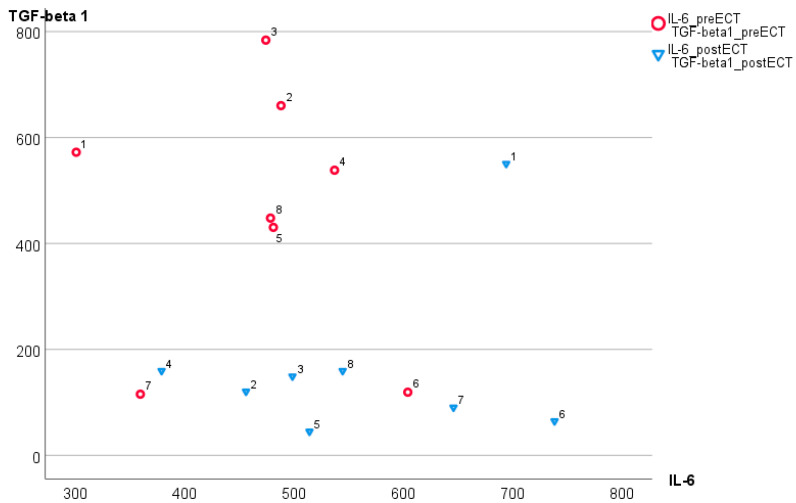
Associations between TGF-β1 and IL-6 before (r = −0.143) and following ECT (r = −0.144) in eight treatment-resistant schizophrenic patients. Numbers are IDs of eight patients.

**Figure 2 biomedicines-12-02637-f002:**
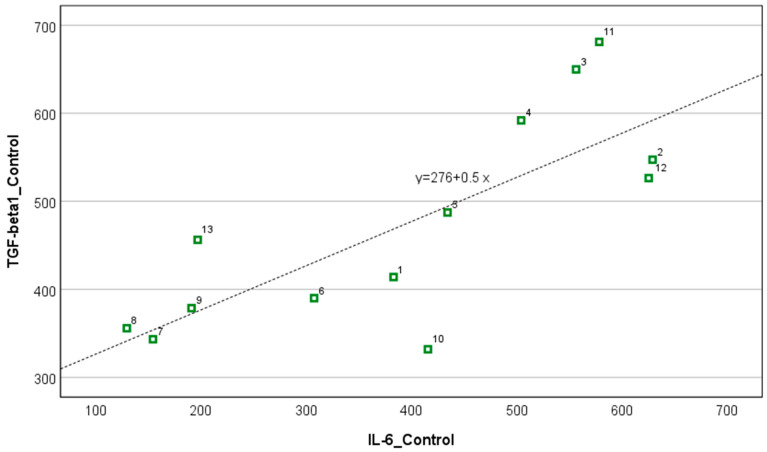
Relationship between IL-6 and TGF-β1 in thirteen control subjects (r = 0.78; *p* = 0.002) Numbers are IDs of thirteen control subjects.

**Figure 3 biomedicines-12-02637-f003:**
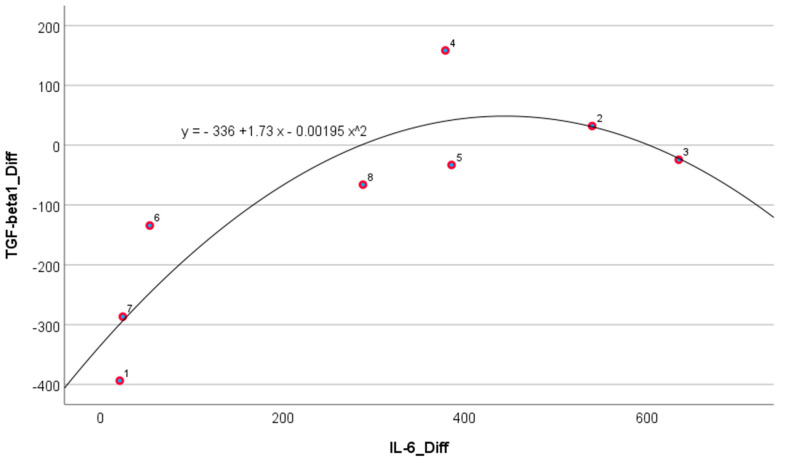
Association between TGF-β1 and IL-6 cytokine changes from pre-ECT to post-ECT in eight treatment-resistant schizophrenic patients (Spearman r = 0.833; *p* = 0.01; R^2^ = 0.789). Numbers are IDs of eight patients.

**Table 1 biomedicines-12-02637-t001:** Patients and treatment characteristics.

No	Sex	Age	Type of Disease	Length of Disease	ECT History	Suicidal Attempts History	SUD #	Clozapine Dose	Chlorpromazine Equivalent of Combined AP	Somatic Comorbidity	PANSS Positive Pre/Post-ECT (% Change)	PANSS Negative Pre/Post-ECT (% Change)	PANSS General Pre/Post-ECT (% Change)	PANSS Total Pre/Post-ECT (% Change)
1.	M	24	R	3	No	Yes	Yes	700 mg	1870 mg	-	17/13 (23.5%)	21/19 (9.5%)	45/28 (28.9%)	83/60 (27.7%)
2.	F	41	P	22	No	No	Yes	150 mg	1150 mg	diabetes, hyperthyroidism	27/18 (33.3%)	33/16 (51.5%)	72/54 (25%)	132/88 (33.3%)
3.	M	20	P	4	Yes	Yes	Yes	-	525 mg	-	25/14 (44%)	33/28 (15.2%)	58/43 (25.9%)	116/85 (26.7%)
4.	M	41	P	20	No	Yes	No	300 mg	675 mg	-	21/17 (19%)	24/16 (33.3%)	58/43 (25.9%)	103/76 (27%)
5.	F	40	R	23	No	No	Yes ^$^	-	635 mg	-	11/9 (18.2%)	27/17 (37%)	52/36 (30.8%)	90/59 (34.4%)
6.	M	41	P	22	No	No	Yes ^$^	750 mg	1875 mg	diabetes, hypertension	24/8 (66.7%)	28/23 (17.9%)	62/43 (30.6%)	114/74 (35.1%)
7.	M	33	P	12	No	No	No	-	325 mg	hypertension	21/12 (42.9%)	33/21 (36.4%)	65/41 (36.9%)	119/74 (37.8%)
8.	F	40	P	5	No	No	No	675 mg	1810 mg	-	25/13 (48%)	33/19 (42.4%)	63/28 (55.6%)	121/60 (50.4%)
										PANSS mean % change	37%	30.4%	32.5%	34.1%

M—Male; F—female; P—schizophrenia paranoid type; R—schizophrenia residual type; SUD #—past substance use disorder; ^$^—Nicotinism only.

**Table 2 biomedicines-12-02637-t002:** PANSS scores pre- and post-ECT in treatment-resistant schizophrenic patients (*n* = 8).

	Pre-ECT Mean (±SD)	Post-ECT Mean (±SD)	Bayesian t-Test for Dependent Variables
PANSS positive symptoms	21.4 (±5.2)	13 (±3.5)	0.002
PANSS negative symptoms	29 (±4.8)	19.9 (±4.1)	0.001
PANSS general	59.4 (±8.2)	39.5 (±8.7)	*p* < 0.001
PANSS total score	109.8 (±16.5)	72 (±11.4)	*p* < 0.001

SD—standard deviation.

**Table 3 biomedicines-12-02637-t003:** Bayesian *t*-test for concentrations of cytokines pre- and post-ECT in schizophrenic patients (*n* = 8) and in control subjects.

Cytokine	Pre-ECT Mean SD	Post-ECT Mean SD	Controls Mean SD	Pre-ECT vs. Controls	Post-ECT vs. Controls	Pre-ECT vs. Post-ECT
IL-6	458.4 238.91	167.4 160.77	148.5 78.92	0.757	0.002 *	0.011 *
IL-12	108.9 45.82	74 15.05	14.8 5.44	0.234	0.006 *	0.05 *
IL-5	11.2 1.02	16.5 10.83	5.6 2.81	0.153	0.822	0.202
IL-10	101.1 38.55	8.7 6.26	473.3 118.09	<0.001 **	0.194	0.001 *
TGF-β1	465.3 94.57	558.8 123.66	291 237.78	0.93	0.124	0.177

Bayesian T—Student test, *p* value; * or **—significance.

**Table 4 biomedicines-12-02637-t004:** The Spearman correlation between cytokine concentrations and the PANSS score. Lower triangle—pre-ECT, and upper triangle (with light gray background)—post-ECT. Dark gray—corresponding variables correlated for measures before and following ECT.

Post-ECT Pre-ECT	IL-6	IL-12	IL-5	IL-10	TGF-β1	PANSS Positive Symptoms	PANSS Negative Symptoms	PANSS General	PANSS Total Score
**IL-6**	0.54	−0.01	0.08	0.44	−0.14	0.57	−0.15	−0.30	0.09
**IL-12**	−0.07	0.17	0.33	0.24	−0.50	0.53	−0.22	0.32	0.48
**IL-5**	0.10	−0.36	0.19	0.12	0.17	−0.01	−0.22	0.06	0.05
**rIL-10**	0.19	−0.21	0.19	0.05	−0.05	0.17	0.21	−0.15	0.16
**TGF-β1**	−0.14	−0.05	−0.14	0.02	−0.26	**−0.79 ***	0.59	−0.49	−0.53
**PANSS positive symptoms**	0.43	0.02	−0.06	0.37	0.24	**0.71 ***	−0.49	0.43	0.69
**PANSS negative symptoms**	0.03	0.01	−0.01	−0.08	−0.06	**0.75 ***	**0.63 ***	−0.11	−0.02
**PANSS general**	−0.18	0.40	0.00	0.08	0.24	**0.72 ***	**0.80 ***	**0.61 ***	**0.88 ****
**PANSS total score**	0.07	0.17	0.12	0.14	0.12	**0.86 ****	**0.91 ****	**0.93 ****	**0.85 ****

*: Correlation is significant at the 0.05 level (2-tailed); **: Correlation is significant at the 0.01 level (2-tailed).

**Table 5 biomedicines-12-02637-t005:** The Spearman correlations of cytokine concentrations in the control group (*n* = 13).

	IL-6	IL-12	IL-5	IL-10	TGF-β1
IL-6	1	0.14	0.48	−0.01	**0.78 ****
IL-12		1	0	0.42	0.15
IL-5			1	0.24	0.47
IL-10				1	0.12
TGF-β1					1

**: Correlation significant at the 0.01 level (2-tailed).

**Table 6 biomedicines-12-02637-t006:** The Spearman correlations between cytokine concentrations and differences in PANSS scores before and after ECT in schizophrenic patients (*n* = 8).

Variable	IL-6 Pre/Post-ECT	IL-12 Pre/Post-ECT	IL-5 Pre/Post-ECT	IL-10 Pre/Post-ECT	TGF-β1 Pre/Post-ECT	PANSS Positive Pre/Post-ECT	PANSS Negative Pre/Post-ECT	PANSS General Pre/Post-ECT	PANSS Total Pre/Post-ECT
IL-6 pre-ECT	0.57	−0.12	0.36	0.31	0.48	−0.08	−0.14	−0.56	−0.40
IL-6 post-ECT	−0.23	−0.06	−0.14	0.54	−0.02	−0.04	−0.24	−0.11	−0.31
IL-12 pre-ECT	−0.24	**0.93 ****	0.02	−0.43	0.00	−0.19	0.13	0.01	−0.04
IL-12 post-ECT	0.45	0.05	−0.24	−0.07	0.43	0.28	**0.87 ****	0.35	**0.74 ***
IL-5 pre-ECT	0.10	−0.40	−0.14	0.24	0.12	−0.39	0.56	0.22	0.20
IL-5 post-ECT	−0.26	0.00	**−0.98 ****	0.52	0.00	0.55	0.46	0.69	0.59
IL-10 pre-ECT	−0.38	−0.10	−0.64	**0.95 ****	−0.26	0.48	−0.08	0.44	0.12
IL-10 post-ECT	−0.33	0.14	−0.24	−0.17	−0.10	0.18	0.02	0.17	0.19
TGF-β1 pre-ECT	0.33	−0.17	−0.52	0.00	0.57	0.22	0.18	−0.10	0.07
TGF-β1 post-ECT	**−0.76 ***	0.19	−0.14	0.33	**−0.93 ****	0.36	−0.40	0.57	0.13
PANSS positive pre-ECT	0.47	0.02	−0.35	0.39	0.39	**0.71 ***	0.44	0.25	0.55
PANSS positive post-ECT	0.49	0.11	0.04	0.08	0.66	−0.15	0.28	−0.37	−0.13
PANSS negative pre-ECT	0.39	0.04	−0.10	−0.05	0.14	0.60	0.63	0.45	**0.84 ****
PANSS negative post-ECT	−0.16	−0.02	0.22	0.08	−0.51	0.62	−0.47	0.16	0.15
PANSS general pre-ECT	0.13	0.37	−0.49	−0.04	0.17	0.53	**0.78 ***	0.61	**0.85 ****
PANSS general post-ECT	0.58	0.36	−0.01	−0.28	0.65	0.21	0.17	−0.32	0.02
PANSS total score pre-ECT	0.33	0.14	−0.38	0.10	0.26	0.58	**0.78 ***	0.53	**0.85 ****
PANSS total score post-ECT	0.51	0.41	0.00	−0.12	0.58	0.29	0.16	−0.27	0.06

*: Correlation is significant at the 0.05 level (2-tailed); **: Correlation is significant at the 0.01 level (2-tailed).

**Table 7 biomedicines-12-02637-t007:** The Spearman correlations between differences in both cytokine concentrations and the PANSS scores in treatment-resistant schizophrenic patients (*n* = 8).

Variable	IL-6 Pre/Post-ECT	IL-12 Pre/Post-ECT	IL-5 Pre/Post-ECT	IL-10 Pre/Post-ECT	TGF-β1 Pre/Post-ECT	PANSS Positive Pre/Post-ECT	PANSS Negative Pre/Post-ECT	PANSS General Pre/Post-ECT	PANSS Total Pre/Post-ECT
IL-6 pre/post-ECT	1	−0.38	0.31	−0.21	0.83 *	−0.01	0.30	−0.51	0.02
IL-12 pre/post-ECT		1	0.07	−0.26	−0.29	−0.06	0.02	0.18	0.01
IL-5 pre/post-ECT			1	−0.48	−0.02	−0.51	−0.36	−0.61	−0.49
IL-10 pre/post-ECT				1	−0.24	0.48	−0.14	0.35	0.07
TGF-β1 pre/post-ECT					1	−0.14	0.35	−0.55	−0.08
PANSS positive pre/post-ECT						1	0.06	0.53	0.59
PANSS negative pre/post-ECT							1	0.48	**0.77 ***
PANSS general pre/post-ECT								1	**0.81 ***
PANSS total score pre/post-ECT									1

*: Correlation is significant at the 0.05 level (2-tailed).

**Table 8 biomedicines-12-02637-t008:** Changes in cytokine, interleukin and neurotrophin concentrations post-ECT vs. pre-ECT in TRS patients.

Study	Number of Participants		Changes in Inflammatory Factors Pre/Post-ECT in TRS Patients									
	TRS	C	BDNF	VEGF	TNF-α	IL-4	NF-kB	TGF-β1	IL-6	IL-12	IL-5	IL-10
Fernandes et al., 2010 [38]	7	21	↔									
Martinotti et al., 2011 [35]	1	0	↑									
Li et al., 2016 [33]	80	77	↑									
Kartalci et al., 2016 [41]	20	20				↑	↔	↑				
Xiao et al., 2018 [36]	40	43		↑								
Ivanov et al., 2019 [39]	66	N/A	↔									
Akbas et al., 2021 [37]	19	35	↔									
Valiuliene et al., 2021 [40]	31	19	↔	↔	↓							
Shahin et al., 2022 [34]	45	N/A	↑									
**This study**	**8**	**13**						↔	↓	↓	↔	↓

BDNF—brain-derived neurotrophic factor; C—control group; ECT—electroconvulsive therapy; Nf-kB—nuclear factor-kB; N/A—not available; TGF-β1 transforming growth factor β1; TNF-α—tumor necrosis factor α; TRS—treatment resistant schizophrenia; ↔ no change; ↓ decreased; ↑ increased.

## Data Availability

The data presented in the study are available on request from the corresponding author(s).

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
