# Peer review of "Changes in IL-6, IL-12, IL-5, IL-10 and TGF-β1 Concentration in Patients with Treatment-Resistant Schizophrenia (TRS) Following Electroconvulsive Therapy (ECT)—A Pilot Study"

_biomedicines, 2024, doi:10.3390/biomedicines12112637_

Round 1

Reviewer 1 Report

Comments and Suggestions for Authors

Authors have performed a pilot study focused on changes in the level of pro- and anti-inflammatory cytokines in eight patients with treatment-resistant schizophrenia following electroconvulsive therapy. The study was carried out in 5 men and 3 women and in thirteen control individuals (2 men and 11 women).

The manuscript is well written, structured and organized. The Abstract summarizes the purpose of the review, and the Introduction covers the literature and explains the basis for the research performed. The Materials and Methods are sound and accurately performed. The inclusion and exclusion criteria are clearly indicated. The Results are also a straight-forward description of the findings observed, and the Discussion is well organized. However, my main concern is the low number, and the heterogeneous population of patients studied.

 Points that must be improved/corrected:

 Authors should consider incorporating a drawing summarizing the main findings reported.

2.      Line 60. Reference 31 has been numbered before 16 and 17. Reference 31 should be number 16.

3.      Check carefully the text. There are some typographical errors.

4.      Results. At the end of this section, a paragraph summarizing the main findings reported is needed.

5.      A comparative Table comparing the findings reported by the authors and those results published in previous studies (for example, references 33 and 34) would help to the reader to have a more complete information on the importance/originality of the findings reported in this small sample of patients with treatment-resistant schizophrenia.

6.      Check carefully the list of references according to the Instructions for Authors.

7.      As indicated above, the article has a major limitation which is the heterogeneity and low number of patients studied. A number much lower than the number of patients studied for example in references 33 and 34. It would be desirable if the authors could increase the number of patients. The recruitment of patients was from December 2018 to February 2020. Why have the authors not continued recruiting more cases after COVID pandemia to increase their number? The authors have previously published other articles in which a larger number of patients were included.

Author Response

Reviewer nr 1

Thank you very much for taking the time to review this manuscript and for you constructive comments and suggestions.  Please find the detailed responses below and the corresponding corrections highlighted in yellow in the re-submitted files.

  1. Authors should consider incorporating a drawing summarizing the main findings reported.

The text summarizing previous studies focusing on the relationship between changes in the immunological system post ECT in comparison to pre-ECT and clinical symptoms in TRS patients has been added (Introduction, lines 110-125).

  1. Line 60. Reference 31 has been numbered before 16 and 17. Reference 31 should be number 16.

The references have been checked again, and correct numeration has been made in the whole text.  

  1. Check carefully the text. There are some typographical errors.

The whole text has been checked  and any typographical errors have been corrected.

  1. Results. At the end of this section, a paragraph summarizing the main findings reported is needed.

A paragraph summarizing the main findings of our research has been added (Results, lines 332-338).

  1. A comparative Table comparing the findings reported by the authors and those results published in previous studies (for example, references 33 and 34) would help to the reader to have a more complete information on the importance/originality of the findings reported in this small sample of patients with treatment-resistant schizophrenia.

 A comparative Table containing the findings reported by the authors and those results published previously has been added in Discussion section, and indicated as Table 7.

  1. Check carefully the list of references according to the Instructions for Authors.

The whole Reference list has been checked according to the Instructions for Authors and additional articles have been added in accordance to the text. .

  1. As indicated above, the article has a major limitation which is the heterogeneity and low number of patients studied. A number much lower than the number of patients studied for example in references 33 and 34. It would be desirable if the authors could increase the number of patients. The recruitment of patients was from December 2018 to February 2020. Why have the authors not continued recruiting more cases after COVID pandemic to increase their number? The authors have previously published other articles in which a larger number of patients were included.

 The Authors agree with both remarks so, they have been included in the Limitation section. Therefore, our research is a pilot study and we have started recruiting patients again, but there are many obstacles to overcome. We would like to point that TRS patients are difficult to study for many reasons, i.e. they do not often agree on ECT treatment  in Poland, sometimes they withdraw from ongoing ECT therapy or they do not give permission to participate in research or they do not agree for blood collection. Thus, the number of patients in the study is small. Therefore, to support the limited number of the patients and to avoid the drawbacks of small samples Bayesian analyses were applied, Material and Methods, lines 212-226.

Kind regards,

The Authors.

Reviewer 2 Report

Comments and Suggestions for Authors

Comments and Suggestions for Authors

Reviewer: While reviewing the manuscript “Changes in IL-6, IL-12, IL-5, IL-10 and TGF-β concentration in patients with treatment-resistant schizophrenia (TRS) following electroconvulsive therapy (ECT) - a pilot study” This study suggests that electroconvulsive therapy (ECT) may have a therapeutic effect in treating patients with drug-resistant schizophrenia (TRS) by adjusting the balance of pro- and anti-inflammatory cytokines. Changes between cytokine levels and psychopathology scores (PANSS) in patients before and after ECT treatment showed a dynamic interaction between the immune system and psychopathology. However, I would like to mention several points that I feel should be addressed before I could recommend this manuscript for publication.

Comment 1: Please explain how electroconvulsive therapy (ECT) works to improve global and positive symptoms in patients with TRS and how it differs from continuing antipsychotic treatment.

Comment 2: Before ECT treatment, IL-10 levels were significantly increased, while IL-5 levels were decreased in TRS patients, which illustrates what results are associated with the inflammatory hypothesis of schizophrenia.

Comment 3: After ECT treatment, IL-6 and IL-12 levels increased, while IL-10 levels decreased in TRS patients, what does this mean, and whether this is related to treatment efficacy.

Comment 4: What is illustrated by the correlation between changes in PANSS scores and changes in cytokine concentrations, does this support the idea that ECT exerts its therapeutic effect by affecting the immune system.

Comment 5: Whether the sample size of this study (8 TRS patients and 13 controls) is sufficient, the reliability of the study results.

Comment 6: In addition to cytokines, ECT may influence the symptoms of schizophrenia through which mechanisms.

Comments on the Quality of English Language

The English could be improved to more clearly express the research.

Author Response

Reviewer nr 2

Thank you very much for taking the time to review this manuscript and for you constructive comments and suggestions. Please find the detailed responses below and the corresponding corrections highlighted in green in the re-submitted files.

Comment nr 1

Please explain how electroconvulsive therapy (ECT) works to improve global and positive symptoms in patients with TRS and how it differs from continuing antipsychotic treatment.

 On the basis of available data, the detailed explanation on how ECT works to improve schizophrenia symptoms and how it differs from continuing antipsychotic treatment has been added in Discussion, lines 511-561 and 585-611.

Comment nr 2

Before ECT treatment, IL-10 levels were significantly increased, while IL-5 levels were decreased in TRS patients, which illustrates what results are associated with the inflammatory hypothesis of schizophrenia.

According to the IRS-CIRS theory of schizophrenia development it was concluded that CIRS plays a key role in the pathophysiology of schizophrenia by negatively regulating the primary IRS and contributing to recovery from the acute phase of illness (Roomruangwong et al., 2020). Therefore, increased concentration of IL-10 pre-ECT in our TRS patients plays a pivotal role in pathophysiology of schizophrenia, not IL-5, which level was decreased.

Comment nr 3

After ECT treatment, IL-6 and IL-12 levels increased, while IL-10 levels decreased in TRS patients, what does this mean, and whether this is related to treatment efficacy

In Abstract section, lines 25-28, the information about increased levels of IL-6 and IL-12 after ECT was incorrect. With regard to our results concentration of pro-inflammatory cytokines: IL-6 (p=0.012), IL-12 (p=0.049) and anti-inflammatory IL-10 (p=0.012) post ECT v. pre-ECT

 were significantly decreased, whereas concentrations of IL-5 and TGF-β1 were not significantly changed. The decrease of IL-6 and IL-12 is related to treatment efficacy.

Thank you very much for noticing the mistake, it should not have happened, but now is correct.

Comment nr 4

What is illustrated by the correlation between changes in PANSS scores and changes in cytokine concentrations, does this support the idea that ECT exerts its therapeutic effect by affecting the immune system.

By examining the correlations between individual interleukins and PANSS scales before and after ECT, the authors wanted to check whether the changes in the concentration of these cytokines are associated with the changes in the severity of schizophrenia symptoms in patients with TRS.

The results obtained by the authors support the idea that at least increase of TGF-β1 concentration induced by ECT could be coupled with attenuation of positive symptoms measured by the PANSS scale. It supports the idea that ECT exerts its therapeutic effect by affecting the immune system (precisely by changes in TGF-β1 level), but further research in bigger groups of TRS patients are required, as the data about involvement of the immune system in therapeutic processes is still scarce. 

Comment nr 5

Whether the sample size of this study (8 TRS patients and 13 controls) is sufficient, the reliability of the study results.

To support the limited number of the patients and to avoid the drawbacks of small samples Bayesian analyses were applied. The use of such an analysis allowed for reliable results, but the authors of the study agree with the Reviewer that conducting these studies on a larger population of patients is fully justified. Materials and Method section, 212-226.

Comment nr 6

In addition to cytokines, ECT may influence the symptoms of schizophrenia through which mechanisms.

The detailed explanation by which mechanisms, apart from cytokines, ECT may influence the symptoms of schizophrenia has been added in Discussion section, lines 511-561.  

Kind regards,

The Authors

Reviewer 3 Report

Comments and Suggestions for Authors

In this article, the effects of electroconvulsive therapy on the cytokine profile of patients with treatment-resistant schizophrenia were evaluated in detail.

Comments to authors:

1.                  In the Discussion, it would be interesting to consider how much metabolic or physiological factors in patients, such as a history of diabetes, might have influenced changes in the cytokine profile.

2.                  Were significant changes in cytokine profile noted between subsequent electroconvulsive therapy sessions?

3.                  How reliable and representative is serum cytokine profile testing compared to cerebrospinal fluid in schizophrenia?

Author Response

Thank you very much for taking the time to review this manuscript and for you constructive comments and suggestions. Please find the detailed responses below. We do believe that our explanations will be found satisfying for you.

Comments to authors:

  1. In the Discussion, it would be interesting to consider how much metabolic or physiological factors in patients, such as a history of diabetes, might have influenced changes in the cytokine profile.

Thank you for your comment and suggestion. We do agree that including information about how diabetes my influence changes in the cytokine profile in schizophrenia patients is interesting, but currently there are not available any data in this matter.

  1. Were significant changes in cytokine profile noted between subsequent electroconvulsive therapy sessions?

            The possible changes in cytokine profile have not been measured due to a lack of written consent for additional blood collection from the patients, which was respected by us. However, we are aware of the fact that concentration of cytokines may fluctuate during ECT therapy.

  1. How reliable and representative is serum cytokine profile testing compared to cerebrospinal fluid in schizophrenia?

 On the basis of current knowledge a straightforward answer on how reliable and representative is serum cytokine profile testing compared to cerebrospinal fluid in schizophrenia is not possible owing to several reasons.

 Firstly, there are the relatively few studies focusing on the level of cytokines in the cerebrospinal fluid (CSF) and they have been limited to a restricted number of cytokines.

For example, Miller and colleagues completed a meta- analysis of cytokine alterations in blood and CSF of patients with schizophrenia (Miller et al., 2011). Blood analysis showed increases in IL-1β, IL-6 and transforming growth factor-β (TGF-β) in acute relapsed patients and first episode patients compared to healthy controls. CSF analysis showed decreased levels of IL-1β in patients compared to controls and no differences in other cytokines. Similar results were found by Upthegrove and colleagues, who found elevated levels pro- inflammatory cytokines IL-1β, sIL-2r, IL-6, and TNF-α in the blood of medication-naive first-episode individuals (Upthegrove et al., 2014). Despite early results by Miller and colleagues indicating a decrease in IL-1β in CSF, a subsequent meta-analysis by Wang and Miller found increased levels of IL-1β, in addition to increased levels of IL-6 and IL-8, in the CSF of patients with schizophrenia (Miller et al., 2011; Wang and Miller, 2018). Research conducted by Gallego et al., 2018 revealed statistically significant increases in levels of IL-8 and IL-1β in the CSF of patients with an schizophrenia spectrum disorder compared to healthy volunteers. Moreover, meta-analysis conducted by them showed statistically significant increases in IL-6 and IL-8 in patients compared to healthy volunteers (Gallego et al., 2018).

Secondly, the research were limited to small numbers of participants and in most of them with a lack of a control group of healthy volunteers.

Thirdly, cytokine levels are likely affected by other genetic, illness, and environmental factors, which were not accounted for in the majority of studies.

Moreover, the sensitivity of the cytokine assays used and appropriate defining the limit of detection (LOD- the smallest concentration of an analyte which can be measured) have huge impact on obtained results (Sighn et al., 2023). Therefore, if there are significant discrepancies in measurement of cytokines concentration within the same used method, the obtained results by different groups of researchers are in comparative.

Therefore, further research in big groups of patients are necessary.

References

Miller BJ, Buckley P, Seabolt W, Mellor A, Kirkpatrick B, 2011 Meta-analysis of cytokine alterations in schizophrenia: clinical status and antipsychotic effects. Biol. Psychiatry 70, 663–671.

Upthegrove R, Manzanares-Teson N, Barnes NM, 2014 Cytokine function in medication-naive first episode psychosis: a systematic review and meta-analysis. Schizophr Res. 155, 101–108.

Wang AK, Miller BJ, 2018 Meta-analysis of Cerebrospinal Fluid Cytokine and Tryptophan Catabolite Alterations in Psychiatric Patients: Comparisons Between Schizophrenia, Bipolar Disorder, and Depression. Schizophr Bull. 44, 75–83.

Gallego JA, McNamara RK, Blanco EA, Castaneda S, Jimenez LD, Alvarez-Lesmes S, Lencz T, Malhotra AK. Evidence for cytokine dysregulation in schizophrenia spectrum disorders: A comparison of cerebrospinal fluid and blood samples. Psychiatry Res. 2024 May;335:115871.

Singh D, Guest PC, Dobrowolny H, Fischbach T, Meyer-Lotz G, Breitling-Ziegler C, Haghikia A, Vielhaber S, Steiner J. Cytokine alterations in CSF and serum samples of patients with a first episode of schizophrenia: results and methodological considerations. Eur Arch Psychiatry Clin Neurosci. 2023 Sep;273(6):1387-1393.

Kind regards,

The Authors

Round 2

Reviewer 1 Report

Comments and Suggestions for Authors

The authors have adequately answered to all the questions raised by the reviewer.